# Influence of Various Climatic Conditions on the Structural Changes of Semicrystalline PLA Spun-Bonded Mulching Nonwovens during Outdoor Composting

**DOI:** 10.3390/polym11030559

**Published:** 2019-03-25

**Authors:** Michał Puchalski, Piotr Siwek, Nicolay Panayotov, Małgorzata Berova, Stanisława Kowalska, Izabella Krucińska

**Affiliations:** 1Centre of Advanced Technologies of Human-Friendly Textiles “Pro Humano Tex”, Department of Material and Commodity Sciences and Textile Metrology, Faculty of Material Technologies and Textile Design, Lodz University of Technology, ul. Zeromskiego 116, 90-924 Lodz, Poland; skowal@p.lodz.pl (S.K.); izabella.krucinska@p.lodz.pl (I.K.); 2University of Agriculture in Krakow, al. Mickiewicza 21, 31-120 Krakow, Poland; p.siwek@urk.edu.pl; 3Agricultural University—Plovdiv, 12 Mendeleev Blvd, 4000 Plovdiv, Bulgaria; nikpan@au-plovdiv.bg (N.P.); maberova@abv.bg (M.B.)

**Keywords:** biopolymers and renewable polymers, polylactide (PLA), degradation, composting, nonwovens, crystallization, crystal form, microstructure evolution

## Abstract

This study analyzed the structural changes of semicrystalline polylactide (PLA) in the form of spun-bonded mulching nonwovens, during outdoor composting. The investigation was carried out at the microstructural, supramolecular and molecular levels using scanning electron microscopy (SEM), wide-angle X-ray diffraction (WAXD) and the viscosity method, respectively. The obtained experimental results revealed how the popular outdoor composting method, realized under two different European climatic conditions (in Poland and in Bulgaria), affects the degradation of PLA nonwoven, designed for agriculture use. The results showed the insignificant influence of the climatic conditions and prepared compost mixtures on the molecular and micromorphological structure of PLA spun-bonded mulching nonwovens, with a visible increase in crystallinity after the first year of composting. Significant changes were observed only after the second year of composting, which indicates the resistance of semicrystalline PLA to degradation in outdoor composting conditions.

## 1. Introduction

In recent years, with regard to environmental preservation and according to the circular economy, novel biodegradable materials, made from biomass waste for application in many areas of life, have been developed [1,2]. A group of biopolymers that has attracted considerable attention includes poly(lactic acid) or polylactide (PLA) [3]. These are synthesized from completely renewable sources, e.g., corn, and possess excellent mechanical properties comparable to those of other polyesters, and can be used for compost or biodegradation [4].

The synthesis of PLA, depending on its application, is based on the polycondensation of lactic acid, or the ring-opening polymerization of lactide (LA) obtained from the depolymerization of oligomers of lactic acid (2-hydroxypropionic) (LAc). LAc is synthesized from hydrocarbons of agricultural origin, such as biomass waste, by means of a fermentation process with the use of bacteria *Lactobacilli*. The polymerization of lactide is usually initiated by covalent alcoholates (Mt(OR)_n_), and results in polylactide composed of macromolecules with ester and hydroxyl terminal groups, respectively [5]. The presence of various stereoisomeric lactide forms, different chirality in the polymer chain, and the creation of different supramolecular structures of polymers strongly influence the physical properties of the final products [6,7,8].

The poly(l-lactide) homopolymer is a crystallizable polymer and can crystallize in three different polymorphous-ordered forms: α [9], β [10], γ [11], one disorder form α’ [12,13], and meso-phase [14] characterized by different properties [15]. In the technological regime, where poly(l-lactide) is usually applied with a low content (<10%) of d-lactide isomers. The main observed crystalline structure form is α’, which ensures the optimum usable properties of final materials [16,17].

PLA is also well known to be hydrolysable and an unstable biodegradable polyester [18,19]. Degradation of this aliphatic polyester depends on the physical, chemical and biological agents, mainly through hydrolytic or thermal degradation [20,21,22]. The degree and rate of degradation also depends on the molecular and supramolecular structures of the polymer [23,24,25]. However, current knowledge is based on the experiments carried out mainly on the amorphous PLA materials such as un-oriented foils or injection molded specimens, degraded under laboratory or natural conditions [26,27]. The presented results from composting tests have demonstrated that materials made from PLA can be degraded after several weeks. However, the temperature of the compost must be near or above the glass transition temperature, which for PLA is in the range of 55–65 °C [28,29]. The degradation of PLA products in outdoor manure piles is slower in a Mediterranean climate [30].

The advantages of PLA have made it a polymer with huge application potential in agriculture in the form of packaging [31], strings [32] and nonwoven fabrics for mulching [33,34]. Due to these potential applications, it is important to carry out an investigation related to the testing of the degradation of prototypes of PLA products during composting under conditions that are typical for agriculture and horticulture—this has become the motivation to carry out the experiment described in this work.

In this study, degradation in an outdoor regime was investigated, which is interesting not only from a scientific point of view, but also from the perspective of polymer materials developed for agriculture use. The possibility of utilizing plastics by composting at home is extremely important with respect to environmental protection, and could be a method of waste reduction. Semicrystalline PLA in the form of spun-bonded nonwovens, such as the material applied to mulching plants in summer 2015, was composted in soil with peat, as well as with and without the addition of a commercially available agent for compost containing nitrogen, namely Radivit^®^, under two different climatic conditions, defined according to Köppen–Geiger’s classification as humid continental (Dfb) and humid subtropical (Cfa) [35]. Additionally, a part of the composting prism was covered by foil which is a typical method used in horticulture and agriculture to accelerate the composting process. The experiment was carried out from 10 April 2015 to 10 October 2017. The effects of degradation were analyzed by using scanning electron microscopy (SEM), wide-angle X-ray diffraction (WAXD) and the viscosity method, which provided information about structural changes at the microstructural, supramolecular and molecular levels, respectively.

## 2. Materials and Methods

### 2.1. Materials

The studied materials were made from the poly(l-lactide) which contained 1.4% of d-lactide, Ingeo™ 6202D (Nature Works LLC, Minnetonka, MN, USA), with the addition of black dye (PolyOne, Avon Lake, OH, USA). The PLA spun-bonded mulching nonwovens with a mass per unit area of 50.1 g/m^2^ were performed using a large laboratory stand. The methodology guaranteed optimal mechanical properties with the previously described semicrystalline structures [36].

### 2.2. Composting Condition and Method

The outdoor composting was realized at two experimental stations: Vegetable Experimental Station of the University of Agriculture in Krakow, Poland (50°04′N, 19°51′E) and Agroecological Centre (AEC) at the Agricultural University of Plovdiv, Bulgaria (42°09′N 24°45′E) where the climate is, according to Köppen–Geiger’s classification, humid continental (Dfb) and humid subtropical (Cfa), respectively [37]. The experiments were started on the last day of June 2014 through the deposition of nonwoven strips on the soil, which were supposed to imitate the litter in contact with the soil. On the last day of August, the litter nonwoven strips were cut into 5 × 5 cm^2^ pieces and added to the compost prisms of the following most popular composting mixtures:Soil with peat (sample notation is defined as S+P)Soil with peat and a commercially available agent for composting, which contains nitrogen, Radivit^®^ (sample notation is defined as S+P+R)

Additionally, part of the compost prisms was covered with foil and that sample is defined as S+P+F and S+P+R+F, respectively. The samples for testing were taken after one year and two years of composting.

### 2.3. SEM Method

The effect of degradation on the change of spun-bonded nonwoven morphology was observed under a scanning electron microscope Nova NanoSEM 230 (FEI Europe B.V., Eindhoven, The Netherlands). The nonwoven samples, after being cleaned with distilled water and dried in a heater at 30 °C, were prepared by fixing the nonwovens to an SEM holder using conducting carbon adhesive tape. The studies were carried out using a Low-Vacuum environment and beam energy 10 keV, which eliminated the requirement to cover the sample with a conductive material such as gold.

### 2.4. Molecular Weight

Molecular weight was measured by the viscosity method in a diluted polymer/dichloromethane (0.08 g/dL) using an Ubbelohde viscometer Type IIa. (SI Analytics GmbH, Mainz, Germany) at 25 °C. The viscosity-average molecular weight, *Mη*, was then calculated from the intrinsic viscosity *[**η**]* using the following equation [38]:(1)$${[\eta ]=KM}_{\eta }{}^{\alpha }$$
where *K* and *α* are constants which equal PLLA 5.45 × 10^−4^ and 0.74, respectively.

### 2.5. DSC Method

The thermal properties of the studied materials were analyzed using a model Q 2000 differential scanning calorimeter, model Q 2000 (TA Instrument Inc., New Castle, DE, USA), in the range of 0–250 °C at a heating rate of 10 °C/min.

### 2.6. WAXD Method

Measurements of the crystalline structure of PLA were carried out with a diffractometer X’Pert PRO (PANalytical B.V., Almelo, The Netherlands) using CuKα source (λ = 0.154 nm) and the following parameters: accelerating voltage 40 kV; anode current 30 mA. A semiconductor counter X’Celector was used as a detector. Diffraction patterns for powdered samples were taken within the range of angles of 2θ: 5°–45°.

## 3. Results and Discussion

### 3.1. Analysis of Micromorphology

In the first part of the evaluation of the degradation of PLA spun-bonded mulching nonwovens, the influence of the degradation factors such as time, climatic conditions and composition of compost on the physical microstructure of the nonwoven fabrics was analyzed. Due to the strong diffusion of the soil inside the nonwoven structure and its difficulty with purification, a reliable assessment of the weight loss was not possible. Thus, in this work, changes at the microstructure level were assessed as an investigation of the physical structures of nonwovens, by means of scanning electron microscopy. Figure 1 presents the SEM images recorded before and after degradation at magnification ×1600. It is clearly shown that the results of the composting process are characteristic transverse cracks of fibers after just one year of degradation [39]. Additionally, in the case of composting in a humid subtropical climate, the preliminary fragmentation of fibers is also observed. After the second year of composting, all of the studied samples are partially fragmented; however, there is a lack of visible fragmentation to the powdered form. Based on the presented results, it can be concluded that the degradation process after two years is still in its early stages at the micromorphology level. The materials have partially fragmented, known as disintegration, without any visible typical effects of fragmentation and mineralization [40]. Moreover, it can also be assumed that changes in micromorphology are slightly more severe when composting is carried out in a humid subtropical climate. In addition, it cannot be determined which compost composition has a more degrading effect on semicrystalline PLA spun-bonded mulching nonwovens based on SEM images of micromorphology (Figure 1). The investigation at the microstructural scale of the effects of degradation also demonstrated an insignificant influence of covering the compost prism with foil on the destruction and fragmentation of the studied samples, mainly visible as surface defects. Covering the compost prism with foil increased the average temperature inside the prism from 25 °C to 32 °C, and from 28 °C to 35 °C in Cracow and Plovdiv, respectively. Based on the assessment of changes in the microstructure of PLA nonwovens, it can be concluded that composting of semicrystalline polymer material under natural conditions is not trivial. Without the proper temperature, soil moisture and natural microorganisms, there are not sufficient conditions for the rapid degradation of the semicrystalline form of PLA.

### 3.2. Molecular Weight Analysis

Figure 2 presents the changes in PLA-estimated intrinsic viscosity and calculated viscosity-average molecular weight that resulted from various composting mixtures and times. The one-year composting process of nonwovens in selected conditions did not significantly change the viscosity-average molecular weight of the PLA. The *Mη* decreased from 38.8 kDa to 37.7–38.0 kDa and 37.0–37.2 kDa in the case of composting in soil with peat alone and composting in soil and peat with Radivit^®^ covered with foil, respectively. The changes in *Mη* are more visible after the second year of composting. Moreover, after two years of the composting process, the influence of climatic conditions on the calculated viscosity-average molecular weight is more visible. According to the presented result, as expected, the most intensive changes in *Mη* parameters are observed in the case of the composting process in the humid subtropical climate (Cfa). After two years of composing, the calculated *Mη* for samples composting in Plovdiv decreased from 38.8 kDa to 31.9–33.0 kDa, while for the samples composting in Cracow *Mη* decreased to only 35.0–34.4 kDa. It is worth noting that the composting process in outdoor conditions—where the maximum temperatures in the composting prism during the experiment were in the case of Cracow in the range of 25–32 °C, and in the case of Plovdiv in the range of 28–35 °C—affected the molecular structure of the polymer, but the effects of degradation are not as significant as in the case of laboratory tests or degradation realized in the municipal composting heap, where the temperatures are in the range of 56–65 °C [29]. The presented results confirm the significant importance of the supplied energy in the form of heat as a degrading factor, which is possible to obtain under natural conditions by changing the climate zone or providing heat from an artificial source. Adding Radivit^®^ agent and covering the composting prism with foil insignificantly accelerate the degradation of semicrystalline PLA material at the molecular level, which is a significant observation from the perspective of PLA degradation in natural conditions. In summary, the obtained results of molecular weight changes correlate with changes at the microstructure level and confirm that after two years of composting PLA nonwoven in natural conditions, degradation is visible but it is still at a preliminary stage.

### 3.3. DSC Results

Analysis of the thermal properties of PLA spun-bonded mulching nonwovens was performed before and after composting in various media and climate conditions.

In Figure 3, the first heating DSC thermograms of the studied materials are presented. Figure 4 compares the estimated variation of the heat capacity (Δ*C*_p_) and degree of crystallinity (χ_c_) for each studied PLA sample. The degree of crystallinity was calculated using the following equation:(2)$$\chi_{c}\;=\;\frac{{\Delta H}_{m}{-\Delta H}_{cc}}{{\Delta H}_{100\%}}\times 100\%$$
where Δ*H*_m_ is the melting enthalpy, Δ*H*_cc_ is the cold crystallization enthalpy and Δ*H*100% is the melting enthalpy of 100% crystalline PLA, which is equal to 93.1 J g^−1^ [41].

On all of the thermograms, the melting point of PLA around 166 °C is clearly visible. The analysis of determined values shows the slight decrease in T_m_ of the studied samples mainly after two years of composting from 166.5 to 166.0 °C. This is an interesting observation because it suggests that most of the crystallites detected by DSC originate from the locally ordered structure obtained during degradation. This phenomenon will be the subject of WAXD studies, the results of which are presented later in this work.

More clearly visible changes in the thermal properties of the studied samples were noted for the glass transition point and cold crystallization peak. For the samples composted in the humid subtropical climate, a decrease in *T*_g_ and *T*_cc_ was discernible, and its range depended on the composting mixture and the presence of a foil cover (Figure 3b). In samples composting for two years, an insignificant or no cold crystallization peak was noted with the significant decrease in the glass transition point. These results correlate with the molecular weight analysis results estimated by the viscosity method. The glass transition point and cold crystallization peak are observed at a lower temperature for the polymer with a lower viscosity-average molecular weight. This phenomenon is evident because the polymer with lower *M*η requires lower thermal energy to transition from a solid to a glassy form. The decrease in molecule weight also results in an insignificant decrease in the cold crystallization point (Figure 3).

The composting process changes the thermal properties of the studied sample. This is mainly observed as a decrease in the values *T*_g_ and *T*_c_, but also changes in the supramolecular structures of the polymer. In Figure 4, the variation in the heat capacity and degree of crystallinity is presented as a function of degradation time. The increase in χ_c_ with decreasing Δ*C*_p_ is clearly presented. According to the presented numerical analysis results of the obtained DSC thermograms, the influence of climatic conditions on the PLA degradation is visible. The increase in the degree of crystallinity with the decreasing variation of the heat capacity is more intense when composting was conducted in a Cfa climate than in a Dfb climate. Moreover, composting with the use of a foil cover contributes to the occurrence of significant structural changes in the polymer in a shorter time, which is clearly visible in the case of samples S+P+F and S+P+R+F. Additionally, the obtained DSC results suggested that the addition of Radivit^®^ to the composting mixture causes an insignificant increase in the velocity of structural changes of PLA during degradation, observed as an increase in the degree of crystallinity with the decreasing variation of the heat capacity, as shown in the viscosity-average molecular weight analysis results.

### 3.4. WAXD Results

In Figure 5, X-ray diffractograms of investigated semicrystalline PLA mulching nonwoven, before and after composting, are compared. In all of the obtained diffractograms, two dominant diffraction peaks located at 2θ, equal to 16.5° and 18.8°, are clearly visible, corresponding to (110)/(200) and (203) lattice planes of α or α’ forms of PLA. In addition, for the composted samples, a sharp diffraction peak at around 2θ, equal to 22°, 27° and 28°, and many weak visible diffraction peaks in the range of 2θ 10°–40°, assigned to the reflection from crystallographic planes of soil, are also discernible.

The visible increase in PLA diffraction peak intensity suggested significant supramolecular changes in the studied polymer. This was analyzed in depth by the deconvolution of the diffractograms into the amorphous halo and the crystalline peaks. For this analysis, the experimental data were fitted by a broad Gaussian peak, characteristic of the amorphous component, and main narrow Gaussian–Lorentzian peaks relevant to the crystalline form, all calculated using the WAXSFIT software based on Hindeleh and Johnson’s method [42]. The shapes of the crystalline peaks and the amorphous halo were matched according to the model proposed by Stoclet et al. [43]. The crystalline phase contents in the studied materials were calculated according to the following equation: (3)$$\chi_{c}=\frac{A_{C}}{A_{C}-A_{A}}\times 100\%$$
where A_A_ and A_C_ are the integral intensities of the amorphous halo and the peaks originating from the crystalline phase, respectively.

Another characteristic feature of the structure formed after composting of semicrystalline PLA spun-bonded nonwoven, under various conditions, was detected by an analysis of the d-spacing (lattice length) of (hkl) crystal planes calculated according to Bragg’s equation [44,45]:(4)$$d_{\left(hkl\right)}=\frac{\lambda }{2\sin\theta }$$
where λ is the wavelength of the X-ray source (0.15418 nm) and θ is an angle of the reflection (half of 2θ of peak position).

Figure 6 presents the numerical analysis results of recorded WAXD diffractograms. The degrees of crystallinity estimated by the WAXD method and d-spacing calculated for the two strong visible diffraction peaks corresponding to (110)/(200) and (203) lattice planes, as a function of degradation time, are similar to those estimated by the DSC method. The degradation of the amorphous phase during composting, along with the ordering of the crystalline phase, caused visible changes in the shape of diffractograms as well as an increase in the determined numerical method of the degree of crystallinity. The influence of the four degradation factors—time, climatic conditions, composting mixtures and covering with foil—on the crystallization of the studied sample is clearly presented. The χ_c_ is higher and d-spacing is lower in the case of composting in a Cfa climate than in a Dfb climate. What is more, covering composting prisms with foil allows faster degradation and crystallization after the first year of conducting the experiment, which is clearly visible in the case of composting under the Dfb climatic condition. Additionally, the addition of Radivit^®^ to the composting mixture causes an insignificant increase in the velocity of structural changes of PLA during degradation, observed as an increase in the degree of crystallinity and decrease in d-spacing. The obtained results testify that the degradation of PLA is not only detectable as an increase in crystallinity but also as the ordering of crystalline forms. For the uncomposted nonwoven, the mean values of the d-spacings were 0.537 nm and 0.474 nm for the (110)/(200) and (203) lattice planes, respectively. The calculated cell units were equal to a = 1.074 nm, b = 0.620 nm, and c = 3.018 nm. For the nonwoven composted in the soil with peat and Radivit^®^ under the Cfa climatic condition, the diameters of the lattice changed, and the mean d-spacings were approximately 0.533 nm and 0.469 nm (a = 1.066 nm, b = 0.615 nm, and c = 2.961 nm). In sum, the initial disorder α’ crystalline form of the studied material is ordering to the α form of PLA during composting.

## 4. Conclusions

An investigation of the structural changes of PLA spun-bonded mulching nonwovens after two years of outdoor composting revealed important information about the degradation of semicrystalline PLA and significantly increased the scientific knowledge about this polymer. The most important conclusions include the following:Degradation of semicrystalline PLA materials by the outdoor composting method in various climatic conditions for one year is not efficient, as confirmed mostly by SEM studies. An increase in the degree of crystallinity with decreasing viscosity-average molecular weight and changes in the thermal properties of the studied material are observed but to a lesser extent than after composting for two years.The climatic conditions and the presence of a foil cover on the composting prism are important for the rapid degradation of semicrystalline PLA by the outdoor composting method. Ensuring high average annual temperatures, preferably close to the glass transition temperature of PLA, and ensuring its stability are crucial from the point of view of composting semicrystalline PLA materials.The addition of a commercially available agent that contains nitrogen (Radivit^®^) to the composting mixture slightly accelerates the degradation process, but not as much as the climatic conditions and covering the composting prism with foil.Outdoor composting had a strong effect on the disorder-to-order phase transition (α’ to α form) of PLA. The decrease in d-spacing measured for the most intense WAXD diffraction peaks is observed after the first year of degradation and is more pronounced after two years of conducting the process. On the basis of the obtained results, it can be assumed that, in the case of PLA degradation, changes in d-spacing and the disorder-to-order phase transition of PLA could be precise degradation assessment indicators.

## Figures and Tables

**Figure 1 polymers-11-00559-f001:**
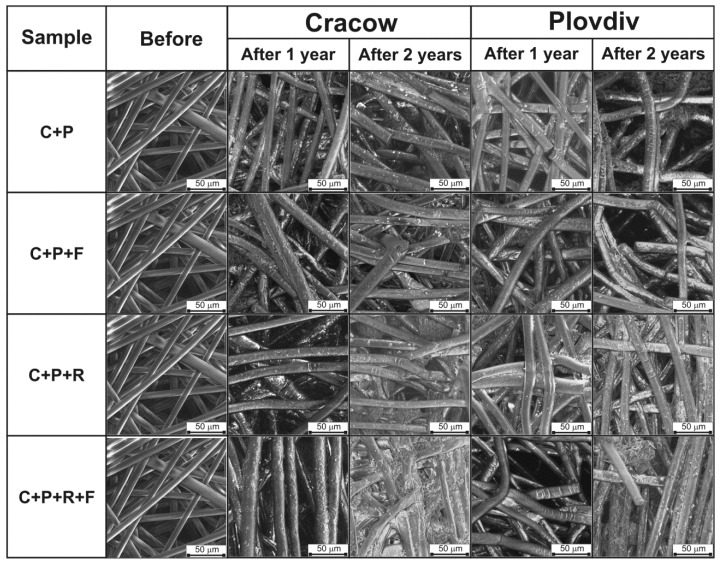
SEM images of PLA spun-bonded nonwovens before and after composting in various conditions.

**Figure 2 polymers-11-00559-f002:**
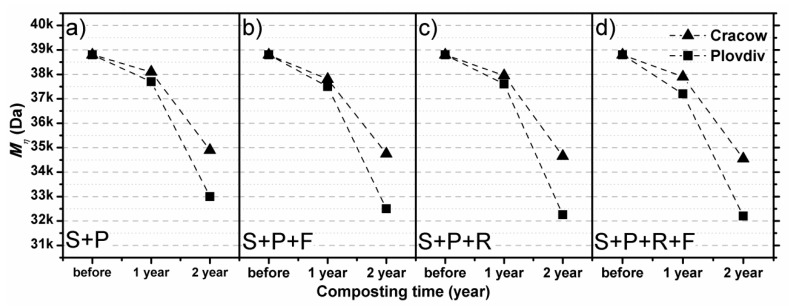
The changes in viscosity-average molecular weight of PLA during composting in: (**a**) soil with peat, (**b**) soil with peat covered with foil, (**c**) soil with peat and Radivit^®^, and (**d**) soil with peat and Radivit^®^ covered with foil.

**Figure 3 polymers-11-00559-f003:**
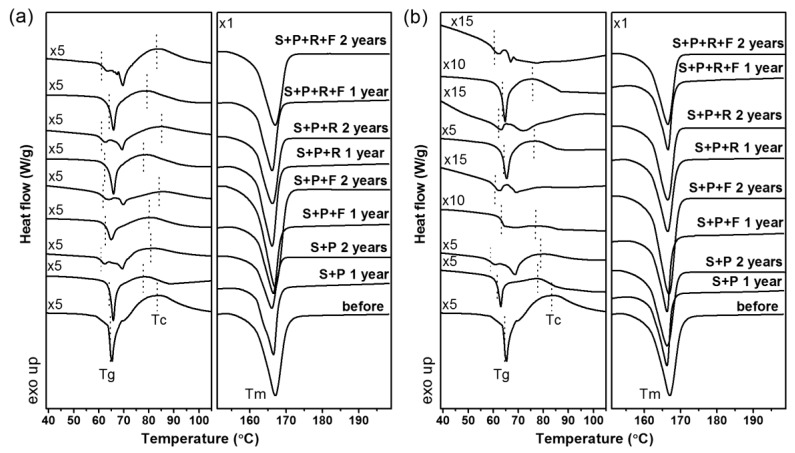
DSC thermographs of PLA spun-bonded mulching nonwovens recorded before and after composting in (**a**) Cracow and (**b**) Plovdiv.

**Figure 4 polymers-11-00559-f004:**
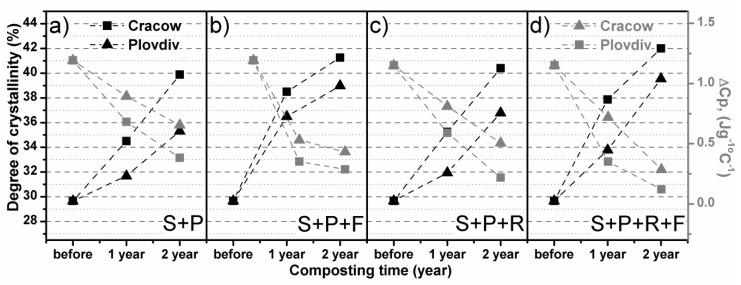
DSC analysis results of semicrystalline PLA spun-bonded mulching nonwovens before and after composting in: (**a**) soil with peat, (**b**) soil with peat covered with foil, (**c**) soil with peat and Radivit^®^, (**d**) soil with peat and Radivit^®^ covered with foil.

**Figure 5 polymers-11-00559-f005:**
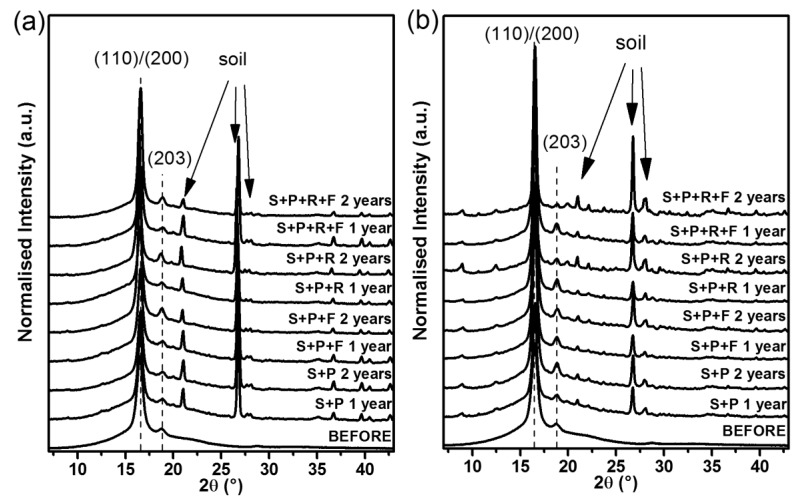
WAXS diffractograms of PLA spun-bonded mulching nonwovens before and after composting in (**a**) Cracow and (**b**) Plovdiv.

**Figure 6 polymers-11-00559-f006:**
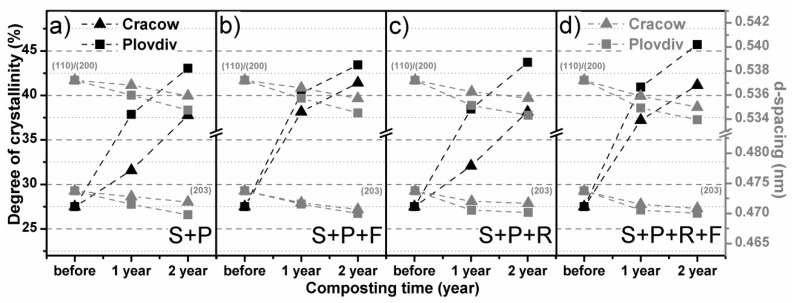
Numerical analysis results of WAXD diffractograms recorded for PLA spun-bonded mulching nonwovens before and after composting in: (**a**) soil with peat, (**b**) soil with peat covered with foil, (**c**) soil with peat and Radivit^®^, (**d**) soil with peat and Radivit^®^ covered with foil.

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
