# Peer review of "Influence of Various Climatic Conditions on the Structural Changes of Semicrystalline PLA Spun-Bonded Mulching Nonwovens during Outdoor Composting"

_polymers, 2019, doi:10.3390/polym11030559_

Round 1
Reviewer 1 Report
It is really a good test for real application, but the most confusing query is what happed during the processing for one year or twe years. The authors should take test for regular sampling, which would be more precise for the experiment. The "sightless" treatment for long time can only be concluded by some results description, without scientific discussion. I can't find the accuracy of influence factors. Also, since some experiments have been carried out mainly on the amorphous PLA materials degraded under laboratory or natural conditions, what's the importance of and why should be semicrystalline PLA researched?
Author Response
The answer on Reviewer Comments
We would like to send the thanks to Reviewer for worth remarks and comments.
We agree with the observation that frequent sampling during the experiment would give even better results in assessing PLA degradation in natural conditions. In the next experiment, we already use material sampling frequently. We have slightly improved the introduction and a correction of the language by the MDPI editor was carried out.
Reviewer 2 Report
The paper deals with the influence of various climatic conditions on the PLA spun-bonded mulching nonwovens. The results show that the degradation of semicrystalline PLA materials in various climatic conditions for one year is not efficient. The discussion is adequate and all justification are substantiated by references. The paper is well structured, it is clear and all conclusions are well supported by the results.
However, the reviewer think there is a mistake. I suggest to revise the Y-symbol "d-sapcing" of Figure 6 because the equation 4 is the crystallite parameters, i.e., the crystallite size of each plane (please refer to some paper, such as Industrial & Engineering Chemistry Research 2013, 52:11996; Journal of Materials Science 2011, 46:7830).
Author Response
The answer on Reviewer Comments
We would like to send the thanks to Reviewer for worth remarks and comments.
We have slightly improved the abstract and the introduction and a correction of the language by the MDPI editor was carried out. Additionally, we decided to the add the suggested reference for the equation 4.
Reviewer 3 Report
In this study the authors evaluated the effect of environmental outdoor conditions on the composting spun-bonded mulching nonwoven PLA. The manuscript does contain valuable information and contribute to the body of the literature which I suggest to be published after revision.
The abstract is not standalone and suffers from lack of data to support the mentioned claims. For example in line 21-23. Those sentences need to be supported with values. The abstract also doesn’t have a conclusion so the reader can easily understand what are exactly the findings of this study. Please revise and add more precise information.
Line 67-68- this sentence is incomplete. “In this study, degradation in an outdoor regime was investigated which is interesting not only 68 from the scientific point of view”…
Page 4/12 line 146: what is the proper temperature, please be specefic
Again line 160-161, same page, what are those most intensive changes in Mn. Please read and revise the whole text for this comment.
Author Response
The answer on Reviewer Comments
We would like to send the thanks to Reviewer for worth remarks and comments.
We have slightly improved the abstract and the introduction and a correction of the language by the MDPI editor was carried out.
Line 67-68- we changed the sentence
Page 4/12 line 146 we added information about extremely change of average temperature
line 160-161,we readed and revised text and belive actual will be correct.
Reviewer 4 Report
This study analysis the structural changes of semicrystalline PLA in the form of spun-bonded mulching nonwoven, during outdoor composting. Despite small English grammar mistakes, it is a well written manuscript with interesting approaches.
Before publication, the authors should provide the details about surface preparation in the SEM analysis and go over the entire manuscript once again to correct the small English mistakes still detected.
Author Response
The answer on Reviewer Comments
We would like to send the thanks to Reviewer for worth remarks and comments.
We have slightly improved the abstract and the introduction and a correction of the language by the MDPI editor was carried out.
We added some details about SEM sample preparation.
Round 2
Reviewer 1 Report
The authors modified the manuscript and answered my queries properly.